# Structure from Silence: Learning Scene Structure from Ambient Sound

**Ziyang Chen**[*], **Xixi Hu**[*], **Andrew Owens**
University of Michigan
https://ificl.github.io/structure-from-silence

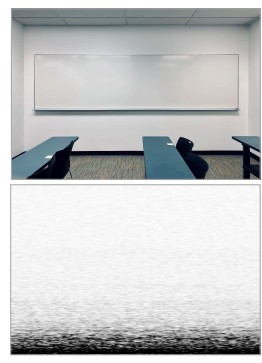
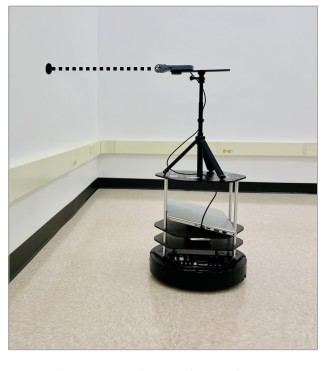
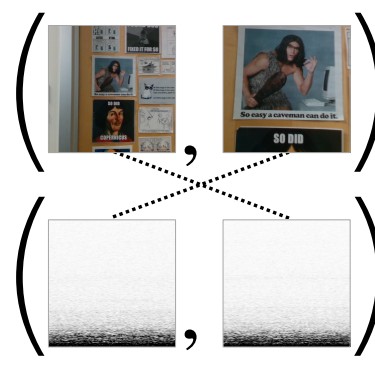

(a) *Quiet Campus* dataset   (b) Depth estimation   (c) Multimodal self-supervision

Figure 1: What can ambient sound tell us about 3D scene structure? (a) We collect an "in-the-wild" dataset of paired audio and RGB-D recordings from quiet indoor scenes. (b) Given audio from a scene, we estimate distance to a wall. (c) We use this ambient sound to learn audio-visual representations through self-supervision.

**Abstract:** From whirling ceiling fans to ticking clocks, the sounds that we hear subtly vary as we move through a scene. We ask whether these ambient sounds convey information about 3D scene structure and, if so, whether they provide a useful learning signal for multimodal models. To study this, we collect a dataset of paired audio and RGB-D recordings from a variety of quiet indoor scenes. We then train models that estimate the distance to nearby walls, given only audio as input. We also use these recordings to learn multimodal representations through self-supervision, by training a network to associate images with their corresponding sounds. These results suggest that ambient sound conveys a surprising amount of information about scene structure, and that it is a useful signal for learning multimodal features.

**Keywords:** audio perception, multi-modal learning, self-supervision, navigation

## 1 Introduction

Humans make extensive use of sound for understanding 3D scene structure. Recent work has sought to reproduce these capabilities in machines, often in simulations with loud, distinctive sounds, such as buzzing alarm clocks and footsteps. Yet it is unclear how often sounds like these are available to robots in practice. Real-world scenes tend to be quiet, and the sounds that we do hear within them, such as ventilation noise, tend to be ambiguous, and propagate through a scene in complex ways.

Human hearing, by contrast, is capable of estimating scene structure from subtle ambient sound cues. Perhaps the most familiar of these are contextual associations, such as knowing that noisy vents and sound-leaking windows tend to be attached to walls. Work in psychology, however, has also proposed that humans use simple low-level cues. These include using the accumulation of low frequencies to tell whether they are near walls [1], using sound shadows to estimate the shape of obstructions [2], and using reverberation to interpret scene geometry [3]. One feature of these cues is that they are available solely through *passive observation*. This is in contrast to echolocation, which requires the observers to actively produce sounds.

---

[*] Indicates equal contribution.

5th Conference on Robot Learning (CoRL 2021), London, UK.

These sounds also provide a "free" learning signal for vision [4]. Learning to successfully predict audio from images (or vice versa) requires understanding how the two modalities vary with pose and scene structure. For example, as one moves toward a sound-making object, it will simultaneously get louder and increase its size within the visual field. Rotating the camera, on the other hand, drastically alters the visual signal without significantly changing the sound. A model that can predict one modality from the other must learn both sources of variation, such as by learning a visual representation that is somewhat invariant to rotations but sensitive to distances to sound sources.

In this paper, we ask whether ambient sounds that occur in *real-world scenes* convey 3D structure, and whether they can be used for multimodal self-supervised learning. To study this, we collected a dataset of "in-the-wild" audio recordings from quiet, indoor scenes typical of what a robot would encounter when solving navigation tasks. Each sound in our dataset is paired with a corresponding recording from an RGB-D sensor, which provides a visual signal and pseudo ground-truth depth. The resulting dataset, which we call the *Quiet Campus* dataset (Fig. 1a), covers a variety of room shapes, background sounds, and materials.

Using this dataset, we conduct an experimental study of depth estimation from audio. First, we show that audio can be used to estimate the distance to nearby walls in a variety of scenarios, including predictions of the relative depth between two recordings. We demonstrate that the model can be used as part of a very simple robotic navigation system, in which a wheeled robot moves along a wall using ambient audio cues (Fig. 1b). Next, we ask whether ambient sound can be used as a learning signal, without explicit use of depth. We study several multimodal self-supervised learning formulations, each of which requires a model to associate subtle changes in sound with a visual signal (Fig. 1c). We show that the resulting model learns a feature set that can be used to solve downstream distance-to-wall estimation tasks.

We make the following contributions: i) a dataset of paired audio and RGB-D data from real-world scenes, ii) an experimental study of audio-based depth estimation, iii) showing that audio-visual recordings from these scenes can provide useful self-supervision for depth estimation tasks. Our results suggest that ambient sound conveys a surprising amount of information about structure, and that it provides a useful training signal for multimodal learning.

## 2   Related Work

**Human auditory perception.**   The field of psychoacoustics has studied what humans can infer from sounds, such as size, material, shape, and depth. One well-known cue for depth estimation is *echolocation*, whereby humans estimate distance by generating sounds and analyzing their echoes [5, 6]. However, humans may also infer a great deal from passive observation. Our work is inspired by Ashmead *et al.*'s studies of the navigation abilities of children with visual impairments [1, 7, 8]. They propose that visually impaired people use accumulations of low-frequency sounds to detect obstacles, and propose physical models of wave interference that would lead to changes in ambient sound fields near obstacles. This effect is similar to how a seashell held to one's ear amplifies ambient sounds to create ocean-like noises [9]. Traer and McDermott [3] show that humans can use phase changes from natural reverberation to distinguish different spaces. While these cues have been studied extensively in psychoacoustics, they are not widely used in robotic perception and self-supervised learning.

**Audio-based depth estimation.**   We take inspiration from work in ocean acoustics that estimates the structure of the ocean from ambient sound using microphone arrays [10, 11]. In classic work, Kac [12] related the shape of a drum to its sound. Recently, Purushwalkam *et al.* [13] reconstructed floor plans from audio-visual signals, using a simulated environment in which sound effects from FreeSound [14] were synthetically inserted into the scenes (*e.g.*, beeping alarm clocks, flushing toilets, and dishwashers). However, it is unclear how often distinctive sounds like these occur in practice, and whether the simplified models of sound propagation are sufficiently accurate. By contrast, we experimentally study audio-based depth estimation in real-world scenes. Other work uses multiple microphones to localize sound sources [15, 16, 17, 18, 19]. Our method, by contrast, requires only monaural audio. Recent work also uses self-produced echolocation sounds produced by onboard speakers. Christensen *et al.* [20] predict depth maps from real-world scenes using echo responses. Gao *et al.* [21] learns visual representations by echolocation in a simulated environment [22]. In contrast, we learn through passive observation, rather than active sensing.

**Audio-visual learning.**   Many recent works have proposed to use paired audio-visual data for representation learning. In seminal work, de Sa [23] proposed multimodal self-supervised learning as an alternative to single-modal models. Later, Ngiam *et al.* [24] learned a multimodal Boltzmann

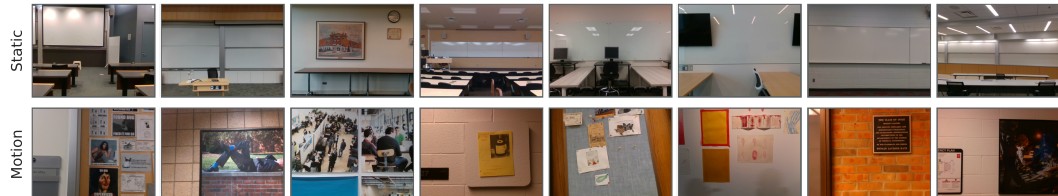

Figure 2: **The *Quiet Campus* Dataset**. We collected a dataset of paired audio and RGB-D recordings from a variety of quiet indoor scenes. We show selected images from the *static* and *motion* subsets, which contain stationary and moving microphones respectively. Please refer to the project webpage for audio-visual examples.

machine. Owens *et al.* [25] learned self-supervised visual representations from impact sounds, and used ambient sound to learn visual features [4]. In the latter, the sounds are taken from internet video and thus contain a much wider range of auditory events than what we consider in this work. Later work simultaneously learned audio and visual representations [26, 27, 28, 29, 30, 31]. Other work has learned cross-modal distillation [32], sound source localization [33, 34, 35, 27, 36, 37, 38, 39, 40], active speaker detection [41, 42, 43], source separation [44, 45, 46, 47]. We take inspiration from this work and show that quiet indoor sounds also provide a useful self-supervised learning signal.

**Visual depth prediction.** Recent work has learned to recover 3D structure from RGB images reconstructions [48, 49, 50, 51, 52, 53, 54], via voxel grids, point clouds, and meshes [55, 56, 57, 58]. Other work has made significant progress predicting relative camera pose [59, 60, 61, 62, 63]. We take inspiration from the techniques proposed in these models, though we predict depth from audio instead of from images.

**Robotic navigation.** Robots often use visual signals to navigate in novel environments [64, 65, 66, 67]. While vision is often a reliable cue for depth estimation, there are many situations where it is unavailable (e.g. due to sensor failures or poor lighting), necessitating "backup" modalities. Recent works have proposed methods that use sound for robot navigation. Those robotic systems are designed to localize sound sources and to navigate to audio goals in indoor environments [68, 69, 22, 70, 71, 72]. Unlike these methods, which largely use distinctive sound sources, we use ambient sounds collected in real-world scenes.

## 3  The *Quiet Campus* Dataset

To study the structure-from-audio problem, we collected a dataset containing a number of "in-the-wild" indoor ambient audio recordings, paired with concurrently recorded RGB-D data. We collected our data from a large number of classrooms and hallways of a college campus. To ensure that the sounds within the recordings were subtle ambient sounds (e.g. ventilation noise), rather than distinctive audio events (e.g. speech), we collected the data at times of the day when there were relatively few people occupying the buildings. Please refer to our webpage for a random sample of sounds.

We collected two types of data: *static* recordings in which the microphone and camera are stationary, and *motion* recordings where a human collector moves slowly toward or away from a wall. We show some visual examples from our collected dataset in Fig. 2. In our experiments, we split the data so that there is no physical overlap, such that the rooms in the training and tests sets are disjoint.

Figure 3: Capturing equipment.

**Static recordings.** We collected data in 46 classrooms from 12 buildings on The University of Michigan's campus, amounting to approximately 200 minutes audio. Inside each classroom, we selected $16 - 30$ positions and recorded 10 secs. of audio at each one. Since we are interested in predicting distance to walls, we point the camera and microphone toward the nearest wall when recording, so that the distance is well-defined.

**Static recordings with dense coverage.** In order to evaluate generalization to different orientations, we use four classrooms from *static* recordings, divided each one into 20 to 35 grid cells of size $1.5 \times 1.5\text{m}^2$ depending on their sizes, and collect 10 hours of audio densely with a wide range of angles (rather than always facing a wall). We call this subset the *static-dense* recordings. These two datasets were recorded at different times of the year. The *static* recordings were made in spring 2021, when buildings were at reduced occupancy due to the COVID-19 pandemic, while *static-dense* were recorded in the winter of 2020, when buildings were at normal occupancy.

**Motion recordings.** To help understand whether small motion (and the subtle changes in audio over space) provides useful information about structures, we collected approximately 90 minutes of videos in motion (222 videos total). To ensure that there are no unintended obstructions nearby, we collect

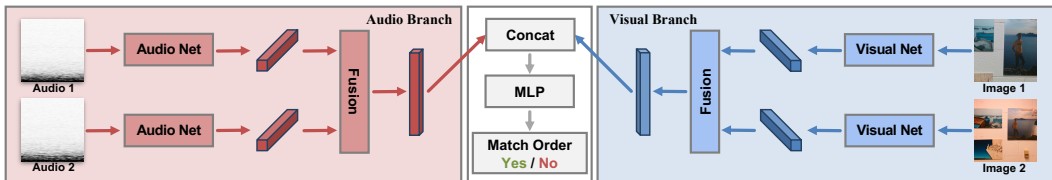

Figure 4: **Self-supervised audio-visual model**. We encode a pair of visual and pair of audio examples using CNNs, and fuse each pair independently using concatenation. The two modalities are then combined and classified using a small multi-layer perceptron, which predicts whether the two pairs are in the same order. Please see the supplementary material for details.

this data in hallways. During recording, the microphone and RGB-D camera move toward or away from a wall (Fig. 2). As in the static setting, we point the microphone toward the wall. To reduce possible sources of sound, the human operator moves slowly and does not take footsteps. We split this dataset based on *building*, *i.e.*, the videos in the training and test set are hallways recorded in different buildings.

**Estimating wall distance.** For experiments that require distance to the wall, we obtain this by cropping the center $320 \times 240$ region from the depth image, and averaging the depth values. Since the camera is oriented toward the wall, this is generally an accurate estimate of depth.

**Hardware.** We use a ZOOM H1n Handy Recorder for collecting audio since it is a simple, inexpensive, and widely-used stereo audio recorder. For the RGB-D data, we use Intel Real-Sense Depth Camera D415 (Fig. 3).

## 4 Predicting 3D Structure from Ambient Sound

We design models and experiments that will experimentally evaluate the ability of a model to learn to predict depth from audio in natural scenes.

### 4.1 Depth estimation tasks

Inspired by work in human obstacle detection with sound [6], we study the ability of a machine hearing system to predict the distance to a wall ahead of them. We pose this problem in multiple ways, which has applications to a variety of robotic navigation applications.

**Obstacle detection.** First, we pose the problem of estimating whether the microphone is within a small distance of a wall (we use 0.5 meters), a binary classification problem.

**Relative depth order.** In many cases, a robotic system needs only to move toward or away from an obstacle, and hence only needs to know which direction is closer to it. We also hypothesize that relative depth estimation is easier task than absolute estimation, as it is in visual depth perception [73, 48]. Given two audio clips, we train a network to predict which one is closer to a wall.

**Relative depth estimation.** Given two audio clips, we train a network to predict the difference between their distances to the wall. We also consider predicting the $\log$ of their difference, following the work in visual depth estimation [50, 49], and as a multi-way classification task, after clustering depth differences in the training set with $k$-means.

**Absolute depth estimation.** Finally, we train a model to regress the distance to a wall, rather than a thresholded "near-or-far" prediction.

### 4.2 Self-supervised audio-visual learning

Having evaluated our model's ability to estimate depth directly from audio, we now ask whether we can learn about structure through audio-visual self-supervision. We hypothesize that, through the task of associating a raw (RGB) visual signal with audio, we will learn a representation with features that will be helpful for downstream depth estimation tasks.

One simple model, considered in prior work, is to solve a audio-visual synchronization task [41, 27, 37]: given the video, distinguish whether audio and visual stream have been misaligned through a random shift. We call this model the **AV-Sync** model. Finally, we pose a new audio-visual learning problem that requires a model to learn *relative* depth between examples. We train a model to distinguish between *ordered* audio-visual pairs, which we call the **AV-Order** model. Given a pair of audio tracks and a pair of images (which correspond to the two audio tracks), we train a network to determine whether the two pairs are in the same (or mismatched) order, which we frame as a classification problem.

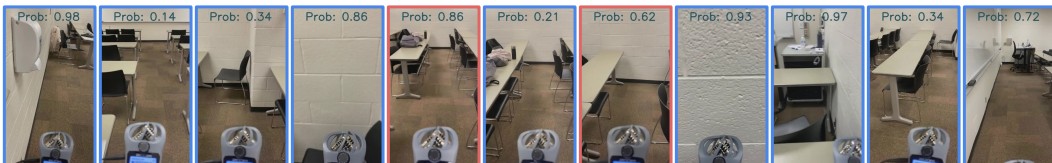

Figure 5: **Obstacle detection**. After training our model, we walk through a room and show the predicted probability of being near a wall. The frames with blue border are correct predictions while frames with red border are failure cases. Please refer the webpage for video results (with audio).

### 4.3 Audio-based robotic navigation

To demonstrate the effectiveness of our depth estimation model, we use it to guide a wheeled robot (TurtleBot) to follow a wall. The robot uses the obstacle detection model to predict whether there is wall near the left/right of the robot, and moves accordingly. We use a simple policy, based on that of Gandhi *et al.* [74] (please see the supplement for full details). The robot moves forward in a sequence of steps. For each step, it rotates to an orientation that is derived from the obstacle detection model. It senses the audio on its left side, then rotates $180°$ and senses audio to its right. We run the obstacle detection model on both audio recordings, and choose an orientation such that the robot moves in the direction that has lower probability of being near a wall.

### 4.4 Models

**Network backbone.** We represent audio using the VGGish network [75]. For our experiments, we either replace the final layer with a new regression or classification head, or we use the (128-dimensional) penultimate layer. The network takes 0.96 sec of audio as input, in the form of a log-mel spectrogram. Due to the importance of low-frequency sound in physical models of wall-sound interaction [8], we do not remove the very lowest frequencies of the audio, in contrast to common practice in speech and audio event recognition [75]. We analyze the importance of this decision through experiments. We convert the stereo audio recordings to mono by averaging the two channels.

For models that use visual data, we use ResNet-18 [76], with $224 \times 224$ images, and modify the last linear layer to output the feature vectors with the same dimension as audio features. We also consider a model whose weights are finetuned from ImageNet [77].

**Learning details.** For fully supervised tasks, we use SGD [78] optimizer with a learning rate $10^{-3}$, momentum 0.9, and weight decay $5 \times 10^{-4}$. For the self-supervised learning experiments, we use Adam [79] with a learning rate of $10^{-4}$ for representation learning. For linear probing experiments, we use SGD with a learning rate of $10^{-2}$. We schedule learning rates via cosine decay and pick the model with best validation performance.

**Audio processing.** Our audio preprocessing resembles that of [75]. We resample the audio to 16 kHz and extract segments of 0.96 secs. We then convert them to spectrograms by applying a Short-time Fourier transform at a stride of 10 ms and 25 ms, and convert them into $96 \times 64$ log-mel spectrograms. For augmentation, we randomly crop the waveform in time and multiply the amplitudes by a random factor uniformly sampled from [0.5, 1.5].

**Depth estimation networks.** For the regression and binary classification tasks, we add a small multi-layer perceptron (MLP) on top of the VGGish network. For the relative estimation tasks, our network uses two audio inputs. For these tasks, we use a Siamese network that fuses the extracted feature vectors via concatenation prior to the MLP. For regression, we normalize depths to be in the range $[-0.5, 0.5]$ during training.

**Network for self-supervised learning tasks.** For AV-Sync task, we concatenate image and audio features before passing through the multi-layer perceptron. For the AV-Order model, we fuse Siamese networks from both modalities via concatenation and add a multi-layer perceptron (Fig. 4). In the downstream task, we take the output features from the last convolution layer after global average pooling and freeze them. We pass our 512-d learned features to a linear layer, which performs classification (and which is the only part of the model optimized during training).

## 5 Experiments

Having collected a dataset of in-the-wild audio and models for inferring structure from it, we conduct an experimental study of what can be inferred from ambient sound in real-world scenes. We report our results along with 95% confidence intervals.

### 5.1 An experimental study of audio-based depth estimation

**Obstacle detection.** First, we evaluate our model's ability to solve the obstacle detection problem (Tab. 1). We split the data such that the test rooms were not heard during training. We report accuracy

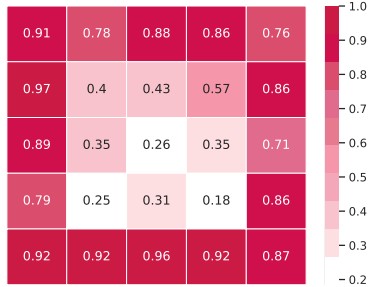

Figure 6: **Wall detection in a single room.** We divide a room into a grid, and record audio within each cell. We show the average obstacle detection probability in each one.

Table 1: **Obstacle detection and relative depth order.** We evaluate our model's ability to determine whether a microphone is within 0.5 meters of a wall and identify which sound has a smaller distance to the wall. *Pre* refers to pretraining.

| Model | Pre. | Task | Obstacle detection AP(%) | Obstacle detection Acc(%) | Relative order AP(%) | Relative order Acc(%) |
|---|---|---|---|---|---|---|
| Audio | | static | 68.3 (±1.3) | 60.0 (±0.9) | 85.5 (±1.0) | 77.2 (±0.8) |
| Image | | static | 99.2 (±0.2) | 95.5 (±0.5) | 94.6 (±0.5) | 86.4 (±0.7) |
| Image | ✓ | static | 99.5 (±0.1) | 98.4 (±0.4) | 97.7 (±0.2) | 92.1 (±0.5) |
| Chance | | static | 46.4 (±1.4) | 50.0 (±1.0) | 47.2 (±1.3) | 50.0 (±1.0) |
| Audio | | motion | 65.6 (±1.4) | 64.5 (±0.9) | 87.1 (±1.0) | 81.3 (±0.8) |
| Image | | motion | 73.4 (±1.2) | 68.2 (±1.0) | 87.9 (±0.9) | 81.2 (±0.8) |
| Image | ✓ | motion | 88.6 (±0.7) | 78.5 (±0.8) | 97.1 (±0.3) | 90.6 (±0.6) |
| Chance | | motion | 50.4 (±1.4) | 50.0 (±1.0) | 50.5 (±1.4) | 50.0 (±1.0) |

on this binary classification task, as well as average precision (to account for possible dataset shift from using held-out rooms). We evaluated our model in both the static and motion settings. To help interpret the effectiveness of audio in the context of other modalities, we also included simple visual models, with both random initialization and ImageNet pretraining[1]. Interestingly, we see that the audio-based model performs better than chance in both cases, suggesting that ambient sound, indeed, conveys information about scene structure.

We also evaluated our model's ability to generalize across orientations and recording times. We found that a model trained on *static* recordings could achieve 86.1% AP and 72.9% accuracy on *static-dense* recordings, indicating strong generalization performance.

We qualitatively demonstrate the obstacle detection results by training a version of the model on the *static-dense* data, since it contains a large number of viewpoint variations and angles (see supplement for quantitative results). We apply the learned obstacle detection model to every frame of a video in which a person walks through a room (Fig. 5). In Fig. 6, we also apply this model densely within a room, divided according to its spatial grid. We averaged the probability of the model for each grid cell. We see that it successfully distinguishes between the boundaries of the room and the center.

**Relative depth order.** In Tab. 1, we evaluate our model's ability to predict which of two audio clips is closer to a wall. We find that the model obtains results that are significantly better than chance, and that relative prediction is significantly easier than obstacle detection. We also

Table 2: **Relative depth ratio.** We evaluate our model's ability of predicting relative depth ratio from two ambient sounds, for the *motion* recordings.

| Model | Regression MAE ↓ | Regression Med. ↓ | Regression $R^2$ ↑ | Regression-by-Classification Top-1 ↑ | Regression-by-Classification Top-5 ↑ | Regression-by-Classification Avg. Dist ↓ |
|---|---|---|---|---|---|---|
| Audio | 0.55 (±.01) | 0.44 (±.01) | 0.48 (±.02) | 22.8 (±0.8) | 80.7 (±0.7) | 1.66 (±.03) |
| Image | 0.54 (±.01) | 0.42 (±.01) | 0.49 (±.02) | 26.6 (±0.8) | 83.6 (±0.7) | 1.47 (±.02) |
| Image (Pre.) | 0.39 (±.01) | 0.29 (±.01) | 0.72 (±.01) | 34.2 (±0.9) | 90.5 (±0.6) | 1.15 (±.02) |
| Chance | 0.89 (±.01) | 0.79 (±.01) | 0.00 (±.00) | 9.45 (±0.6) | 52.6 (±1.0) | 2.79 (±.04) |
| No input | 0.82 (±.01) | 0.75 (±.01) | 0.00 (±.00) | 10.7 (±0.6) | 51.6 (±1.0) | 4.50 (±.05) |

asked whether our model might use binaural cues (even though the stereo channels are averaged). Using only the left channel, we obtain nearly identical performance: 87.2% AP and 81.2% accuracy. Finally, we analyzed the effects of mixing in spurious distractor sounds (see supplement).

**Relative depth ratio.** Since the network is able to tell relative depth order well, we explore relative depth setup further by predicting depth difference of audio pairs (log depth ratio). We use regression to predict the log-ratio from ambient sounds with an L1 loss and via classification. We report the performance of mean absolute error, median error, and $R^2$. For classification, we report top-1 and top-5 accuracy, and report average distance from the ground truth bin using the distance between centers of paired bins. We report the performance of

Table 3: **Absolute depth estimation.** We evaluate our model's ability of predicting absolute distance to the wall for the *motion* recordings.

| | Model | Regression MAE ↓ | Regression Med. ↓ | Regression $R^2$ ↑ | Regression-by-Classification Top-1 ↑ | Regression-by-Classification Top-5 ↑ | Regression-by-Classification Avg. Dist ↓ |
|---|---|---|---|---|---|---|---|
| Single | Audio | 0.28 (±.00) | 0.25 (±.01) | -0.34 (±.03) | 30.8 (±0.9) | 88.3 (±0.6) | 1.11 (±.02) |
| | Image | 0.31 (±.00) | 0.27 (±.01) | -0.67 (±.07) | 35.6 (±0.9) | 95.9 (±0.4) | 1.05 (±.02) |
| | Image (Pre.) | 0.26 (±.00) | 0.21 (±.01) | -0.24 (±.04) | 50.8 (±1.0) | 99.2 (±0.2) | 0.62 (±.01) |
| | No input | 0.28 (±.00) | 0.27 (±.01) | -0.19 (±.02) | 24.3 (±0.8) | 88.3 (±0.6) | 1.07 (±.01) |
| Conditional | Audio | 0.21 (±.00) | 0.17 (±.00) | 0.19 (±.02) | 36.9 (±1.0) | 90.0 (±.06) | 1.17 (±.02) |
| | Image | 0.22 (±.00) | 0.18 (±.00) | 0.12 (±.02) | 38.2 (±0.9) | 95.5 (±0.4) | 0.93 (±.02) |
| | Image (Pre.) | 0.18 (±.00) | 0.14 (±.00) | 0.39 (±.02) | 51.7 (±1.0) | 99.8 (±0.1) | 0.59 (±.01) |
| | No input | 0.25 (±.00) | 0.23 (±.00) | 0.01 (±.01) | 26.4 (±0.8) | 95.9 (±0.4) | 1.43 (±.03) |
| | Chance | 0.78 (±.01) | 0.84 (±.01) | -3.38 (±0.23) | 23.3 (±1.2) | 56.9 (±0.9) | 2.83 (±.02) |

---

[1] We do not use a pretrained VGGish audio model since existing models remove low frequencies and are not compatible without significant changes.

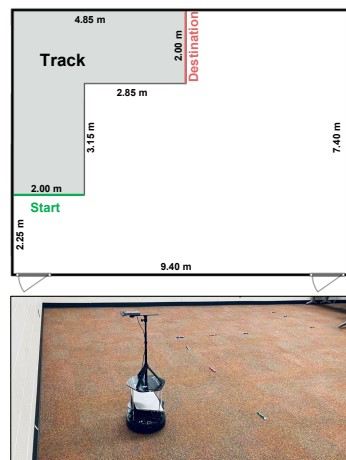

Figure 7: Classroom floor plan and track setting.

Table 4: **Linear probing experiments.** We evaluate our self-supervised feature set for **obstacle detection** and **relative depth order**, for the *motion* recordings. Here, *Audio* means taking audio only as inputs. *Visual* means taking images only as inputs. *Both* means taking both audio and image as inputs.

| | Model | Pre. | Obstacle detection AP(%) | Obstacle detection Acc(%) | Relative order AP(%) | Relative order Acc(%) |
|---|---|---|---|---|---|---|
| Audio | Scratch | | 61.9(±1.5) | 60.3(±0.9) | 78.0(±1.4) | 73.1(±0.9) |
| | VGGish [75] | | 58.2(±1.3) | 56.0(±1.0) | 61.1(±1.4) | 61.2(±1.0) |
| | AV-Sync | | **69.1**(±1.4) | **64.0**(±0.9) | 80.2(±1.3) | 74.1(±0.8) |
| | AV-Order | | 63.4(±1.4) | 61.5(±0.9) | **84.2**(±1.2) | **79.4**(±0.7) |
| | VGGish [75] | ✓ | 59.0(±1.5) | 56.7(±1.0) | 67.7(±1.4) | 64.5(±0.9) |
| | AV-Sync | ✓ | **65.3**(±1.4) | **62.8**(±0.9) | 82.1(±1.2) | 76.4(±0.8) |
| | AV-Order | ✓ | 62.8(±1.5) | 64.5(±0.9) | **85.5**(±1.1) | **80.7**(±0.8) |
| Visual | Scratch | | 70.1(±1.3) | 64.0(±0.9) | 79.7(±1.1) | 71.5(±0.9) |
| | AV-Sync | | **77.1**(±1.1) | **69.2**(±0.9) | 85.3(±0.9) | 76.1(±0.8) |
| | AV-Order | | 76.8(±1.1) | 68.8(±0.9) | **87.4**(±0.9) | **79.1**(±0.8) |
| | ImageNet [77, 76] | ✓ | 80.4(±1.2) | 74.5(±0.8) | 94.0(±0.5) | 85.8(±0.7) |
| | AV-Sync | ✓ | **89.0**(±0.8) | 75.6(±0.8) | 92.8(±0.6) | 85.4(±0.7) |
| | AV-Order | ✓ | 86.5(±1.1) | **76.3**(±0.8) | **95.8**(±0.4) | **88.9**(±0.6) |
| Both | AV-Order | | 77.1(±1.1) | 69.1(±0.9) | 89.0(±0.8) | 80.8(±0.8) |
| | AV-Order | ✓ | 88.1(±0.9) | 76.9(±0.8) | 95.8(±0.4) | 88.9(±0.6) |

zeroing out two inputs as a baseline. As shown in Tab. 2, we can see that our method outperforms chance and, interestingly, has a relatively small gap between the vision-based approach.

**Absolute depth estimation.** We evaluated our model's ability to predict depth from a single audio example, using similar experiments as the relative depth ratio prediction (Tab. 3). Our model is more accurate than chance (which we obtain for all metrics by simply ablating the input and training). Likewise, the regression-by-classification model improves over chance performance.

We also consider another model that estimates depth for a given audio input after conditioning on both a *reference sound* recorded within the same scene and its depth (see the supplementary material for details). We call this the *conditional* absolute depth model. This model obtains better performance than our unconditional model, suggesting that the reference audio provides useful information.

## 5.2 Audio-visual representation learning

Finally, we investigate whether we learn useful representation with **AV-Order** and **AV-Sync** models. In the AV-Order binary classification pretext task, our model obtains 85.8% AP and 75.8% accuracy (80.3% AP and 70.7% accuracy with random initialization) on the test set where the random chance is 50%. Given its performance on the pretext task, we test whether the learned audio and visual representations are useful for the downstream depth-estimation tasks: (1) obstacle detection and (2) relative depth estimation. To evaluate the quality of our spatial representation, we freeze the learned features and train a linear classifier on the depth estimation tasks (Section 4.1). In the downstream tasks, we evaluate image and audio features individually and compare them with random features, pre-trained ImageNet features [77, 76], pre-trained AudioSet features [75] and **AV-Sync** features. Since AudioSet pre-trained weights require a special audio pre-processing that differs from ours, we provide another random feature baseline based on the same processing as [75].

Table 5: **Linear probing experiments.** We evaluate our learned representation for **relative depth ratio** for the *motion* recordings.

| | Model | Pre. | Top-1 (%) ↑ | Top-5 (%) ↑ | Avg. Dist ↓ |
|---|---|---|---|---|---|
| Audio | Scratch | | 19.2(±0.8) | 72.8(±0.8) | 2.33(±0.04) |
| | VGGish [75] | | 14.4(±0.7) | 53.9(±1.0) | 3.78(±0.06) |
| | AV-Sync. | | 19.2(±0.7) | 72.7(±0.8) | 2.07(±0.03) |
| | AV-Order | | **22.2**(±0.9) | **79.6**(±0.8) | **1.86**(±0.03) |
| | VGGish [75] | ✓ | 15.6(±0.7) | 54.0(±1.0) | 3.59(±0.05) |
| | AV-Sync. | ✓ | 20.7(±0.8) | 75.1(±0.9) | 1.99(±0.03) |
| | AV-Order | ✓ | **23.6**(±0.9) | **80.5**(±0.7) | **1.75**(±0.03) |
| Visual | Scratch | | 18.5(±0.8) | 70.8(±0.8) | 2.66(±0.05) |
| | AV-Sync. | | 22.2(±0.8) | 76.8(±0.8) | 1.85(±0.03) |
| | AV-Order | | **24.7**(±0.8) | **80.2**(±0.8) | **1.71**(±0.03) |
| | ImageNet [77, 76] | ✓ | 27.4(±0.9) | 87.1(±0.7) | 1.60(±0.03) |
| | AV-Sync. | ✓ | 27.5(±0.8) | 85.2(±0.7) | 1.53(±0.03) |
| | AV-Order | ✓ | **28.9**(±0.9) | **88.6**(±0.6) | **1.40**(±0.03) |
| Both | AV-Order | | 23.8(±0.8) | 81.5(±0.7) | 1.59(±0.03) |
| | AV-Order | ✓ | 30.0(±0.9) | 89.3(±0.6) | 1.31(±0.03) |

**Relative depth.** We report results for relative depth order and relative depth ratio in Tab. 4 and Tab. 5 respectively. Our method outperforms baselines in both audio and visual modalities suggesting that we learn useful depth representations. Moreover, our results are often close to supervised features.

**Obstacle detection.** As the results shown in Tab. 4, AV-Order model outperforms Scratch and AudioSet pretraining, while it loses to AV-Sync model. The AV-Order obtains worse performance since AV-Order, perhaps because the model deals with two inputs during the pretext task. We provide

an additional experiment by fine-tuning our learned AV-Order feature and we compare with the fully supervised method in Tab. 1. This obtains a 4% AP boost (70.0% v.s. 65.6%), suggesting that our learned representation is helpful.

Interestingly, the multimodal models that use *both* audio and visual features consistently outperform the single-modality models, suggesting the audio is useful in conjunction with a visual signal. We conduct few-shot learning experiments to show audio-visual self-supervision improves fine-tuning performance (please see the supplementary for the details).

**Audio cues.** To help understand what information the model is using, we retrained our model with access to only part of the frequency spectrum. We zeroed out different frequency ranges in the Mel spectrogram, then re-trained the relative depth order model. As can be seen in Fig. 8, the accuracy of the model with low frequencies (0-1000Hz) is comparable to the model with the full frequency range, suggesting that our model mainly use those frequencies to infer the structure information. This may be because there is relatively little high frequency information in the dataset. We also compare the data distribution of our dataset with (non-ambient) music sounds from the VGGSound dataset [80] (see supp. for details).

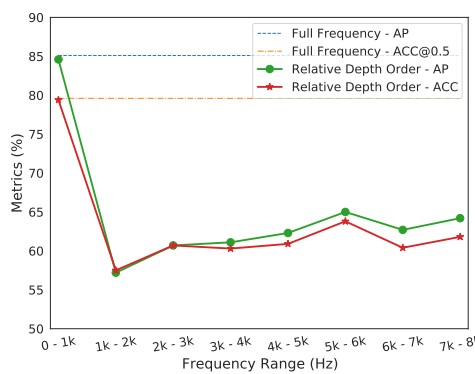

Figure 8: Performance vs. different frequency.

### 5.3 Robotic navigation with ambient sound

We ask whether our depth estimation model can be used to guide a robot to follow a wall, solely using ambient sound. For this, we used a model trained on *static* recordings, and evaluate on a classroom that was not included in the training set. The track is 8m long, and forces the robot to turn a corner (Fig. 7). The robot starts navigating from one of several unknown locations (40cm, 80cm, or 120cm from the wall) and orientations (30° left or right, or facing forward). We repeat the experiment 18 times and measure the average distance along the track it attained before colliding with a boundary. Quantitative results are shown

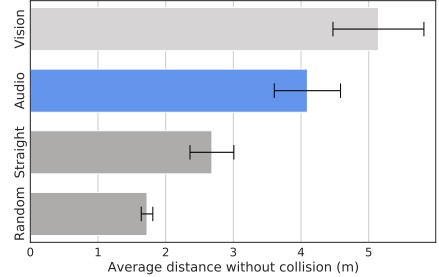

Figure 9: Robot navigation results.

in Fig. 9. We compare with a random policy, a "straight line" policy that always drives forward, and a policy based on the vision model. For the random policy, the setting is the same as our method, except that we use an obstacle detection network with random weights. The straight line policy is effective when the robot starts with a front-facing orientation. It also has the advantage that it does not rotate to sense audio, and hence avoids accumulating errors[2]. As expected, the visual model significantly outperforms the audio model. We see that our method performs better than random and straight-line policies, and successfully guides the robot past the corner in nearly half of the trials. We repeat experiments in the different rooms (please see the supplementary for additional results).

## 6 Conclusion

We see this work as opening two directions. The first is in creating robotic systems that exploit real-world, passively obtained audio signals for depth estimation, especially as an inexpensive "backup" signal that can be used when other modalities fail. Second, we see our work as a step toward using inexpensive and plentiful audio signals to help train other modalities, such as vision.

**Limitations.** While ambient sound provides useful information about scene structure, it is generally less effective than vision. We also lack a full understanding of what cues the model is using, and what can (and cannot) be perceived from it, such as whether there are fundamental limitations on resolution, material, or angle. Additionally, our experiments were conducted on a single college campus, and hence may not be representative of all indoor spaces.

---

[2]These errors could be reduced by using a multi-microphone array, though for simplicity we use a single microphone.

**Acknowledgments**

We thank James Traer for his invaluable help explaining work on human perception of ambient sound. We would also like to thank Linyi Jin and Shengyi Qian for help with visual experiment setups and their valuable suggestions. We also thank David Fouhey for his comments and feedback, Peter Gaskell for his help on robot equipment, and Yichen Yang for the help on data collection. We thank the valuable suggestions and feedbacks from David Harwath, Kristen Grauman and UT-Austin Computer Vision Group. This work was funded in part by DARPA Semafor and Cisco Systems. The views, opinions and/or findings expressed are those of the authors and should not be interpreted as representing the official views or policies of the Department of Defense or the U.S. Government.

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
