# OpenReview forum: "Structure from Silence: Learning Scene Structure from Ambient Sound"
_robot-learning.org/CoRL/2021/Conference — CoRL2021 Oral_

### Official Review · Reviewer_jRS5 · 2021-07-20

**Originality:** Very Good
**Technical Quality:** Very Good
**Clarity Of Presentation:** Fair
**Impact:** 4

**Recommendation:**

Weak Accept: I recommend accepting the paper, but will not argue for my recommendation if the majority of other reviewers have a different opinion.

**Summary:**

This paper explores the problem of deploying a system able to estimate depth from ambient sounds. The paper also shows how to combine ambient sound with image in a self-supervised learning task.

**Issues:**

- The paper contains too much information and details and sometimes it is difficult to follow. The template for this conference does not provide much space so too many small subsections are not a good idea because the information is scattered. Several sections can be merged, for example, subsection "Evaluating the representation." into "Experiments," section "4.1" into "Experiments", etc.

- Experimental results are sometimes unclear with respect to the input: audio, image, both? For example, Tables 4 and 5 separate the audio from the video but what I understood from the paper is that both are combined in the input as shown in Figure 4. Moreover, the link/relation between section 4.2 and section 5.2 is somehow unclear to me.

- When estimating the depth in Tables 1 and 2, it makes sense to compare with random but I do not see the point of comparing with images alone. The paper is about sound. Is vision the baseline?

- In section 5.4, authors can add a comparison to a robot that uses images to estimate depth.

Two final thoughts:
- I like the paper as a proof of concept, and I am curious to find some potential direct application of this system. Any comments on this?
- What happens when the environment contains other non-ambient sounds?


**Reviewer Expertise:**

Fair: Some knowledge of the area

**Strengths And Weaknesses:**


*** Strengths:
+ The idea seems original

+ Experimental results are thorough

+ A good dataset is provided.


** Weaknesses: See Issues

**Summary Of Recommendation:**

The idea seems original and the paper provides numerous experiments together with a dataset that can be useful to other researchers. Therefore, I initially suggest acceptance.

---

> ### Author Response · Authors · 2021-08-30
> **Response to Reviewer jRS5**
>
> Dear Reviewer,
>
> Thanks for your valuable and helpful comments. We greatly appreciate that you found the work original and thorough. We address your questions and suggestions as follows:
>
> **Q4.1: Clarity Of Presentation**
>
> A4.1: Thanks for the valuable suggestions. We have reorganized the paper based on your suggestions, which indeed simplifies the exposition. There are now fewer subsections. Please see the revision.
>
> **Q4.2: Clarification of model inputs in Table 4 & 5**
>
> A4.2: Thanks for the suggestion. We have added more explanation of the tables in the revised paper to clarify this. To summarize: Figure 4 shows the overview of the self-supervision task, in which the model takes both images and audio as input. Table 4 & 5 show the results of downstream tasks using our self-supervised model, so the input is either pretrained image or audio features.
>
> **Q4.3: Why comparing with image model in Table 1 and 2?**
>
> A4.3: The reason we provided the visual model was simply to give a reference point that makes it easier to interpret the results of our audio and audio-visual models. We feel that this helps a reader interpret the quality of the audio results. We view the image model as an approximate upper bound of these tasks, since we have known that vision is effective for solving depth estimation.
>
> **Q4.4: Robot experiment using vision model**
>
> A4.4: We followed this suggestion and ran a visual model on the robot during the rebuttal period. While we have no doubt that a visual model could excel at this task (e.g. using SLAM), our goal was to keep the model as close as possible to the audio controller. We made small modifications to Algorithm 1 in the supplementary to accommodate vision modality, and trained our visual model on additional data with a wider variety of angles to the wall (since in early experiments we found the visual model to be sensitive to orientation). The details of the algorithm we used for the vision model are provided in the updated version of the supplementary material. Please refer to Figure 9 in the updated paper (also can be seen [here](https://drive.google.com/uc?export=view&id=1SGdnpL_VHhjsyQ8EDA9lKOZ6YKKXgo4X)) for the results of the vision model. The vision model performs better than the audio model, and the average distance without collision is 1m further than the audio model.
>
> **Q4.5: Potential direct applications**
>
> A4.5: While we consider our main contribution to be the scientific study of ambient sound for scene structure estimation, there are a number of important applications, especially in *multi-modal* robotics systems. 1) One of the promises of multi-modal perception is *robustness to failures*. There are many cases when vision is unavailable, such as when lights in a room are turned off, a lens is occluded, a camera fails, or there is a networking failure. Audio is an inexpensive signal that is often already available with cameras, and our experiments suggest it could be useful in a "backup" role, much like it is for humans who lose their sense of sight. 2) Audio provides a useful signal for *self-supervised training* of other sensory modalities. Our experiments suggest that visual models trained with audio are more effective than those trained from scratch (Table 4, Table 5). 3) Audio may be a useful extra cue, as part of multi-modal models (see our new experiments in A1.4, which show an improvement with multi-modal features over single-modal features).
>
> **Q4.6: The effect of non-ambient sounds**
>
> A4.6: We'd like to emphasize that our goal is to evaluate whether *ambient* sound is useful for learning scene structure. Previous work has already shown that non-ambient environmental sounds provide important cues for navigation  [11, 12, 13], so if these sounds appeared in our dataset, they would confound our experiments: it would be unclear which sound (novel ambient or previously-studied non-ambient) our model was using. To answer the reviewer's question while avoiding these sounds from providing useful information to the model, we made a synthetic dataset by mixing our recorded dataset with random sounds from FreeSound [68] and evaluating on the obstacle detection task. Note that FreeSound itself might include some ambient sound, which would distract the model. Please refer to Section G and Figure 13 in our updated supplementary (Figure 13 can also be seen [here](https://drive.google.com/uc?export=view&id=1_8CZ0LybUlHPNCy2awICsCJjsUVKX4-D)). In Figure 13. "Retraining" is a version of the model that has been trained on this new dataset. It can be seen that our model can still learn to solve the obstacle detection task, despite uninformative non-ambient sounds.

---

> > ### Comment · Reviewer_jRS5 · 2021-09-03
> > **Rebuttal**
> >
> > OK.

---

### Official Review · Reviewer_13Bf · 2021-07-23

**Originality:** Good
**Technical Quality:** Good
**Clarity Of Presentation:** Good
**Impact:** 4

**Recommendation:**

Strong Accept: I recommend accepting the paper and will argue for my recommendation even if other reviewers hold a different opinion.

**Summary:**

* This paper shows that ambient sound conveys a surprising amount of information about scene structure. This information can be successfully exploited by learning systems.
* Introduces the quiet campus dataset: "in the wild" audio recordings paired with RGB-D data from quiet indoor scenes.
* Experimental study for depth estimation from audio.
* Results showing that ambient sound can be used to infer depth-aware visual representations.


**Issues:**

Update: my early concerns have been addressed and I would like to raise my score.

**Reviewer Expertise:**

Very good: Comprehensive knowledge of the area

**Strengths And Weaknesses:**

# Strengths
* Interesting motivation: humans use subtle spatial variations in the ambient sound spectrum to better perceive 3D scene structure. An important open question is whether this same information can be used by machines for perception or as a labeling source for self-supervised cross-modal learning.
* Data collection. good variation in data collected that represent realistic scenarios. Static recordings, recordings under motion. Running experiments on cheap commodity hardware will help reproducibility.
* Diverse set of depth estimation tasks with different motivations that are relevant to multiple robotic learning problems.
* Good details on training visual-audio self-supervised representations that will be useful beyond this work.

# Weaknesses
* Claims of significance without providing statistically significant evidence. All experiments are presented as point estimates without standard deviation over multiple training runs. Are these numbers the maximum over a large hyperparameter sweep of a single model or the average expected performance? Without this information, it is difficult to substantiate whether the main claims of the paper are indeed true or if we are simply observing sampling error. Examples:
L226: "Interestingly, we see that the audio-based model performs significantly better than chance in both cases, suggesting that ambient sound, indeed, conveys information about scene structure".
L241: "Once again, we find that the model obtains significantly better results than chance"
L262: "As shown in Tab. 2, we can see that our method significantly outperforms chance and, interestingly, has a relatively small gap between the vision-based approach" Hard to say "significantly" or "relatively small gap" without confidence intervals.
L266
L294: "Our method outperforms baselines significantly in both audio and visual modalities suggesting that we learn useful depth representations. Moreover, our results are often close to supervised features". This is a very strong statement to make on pretty weak evidence. For example, the point estimate success between pretrained Imagenet is 27.4 and your AV-Order success is 28.9. Is a 1.5% gap actually significant? It's difficult to say without confidence intervals.


**Summary Of Recommendation:**

This work motivates ambient sound as a rich source for self-supervised cross-modal learning. It presents early experiments that show this new kind of data can be exploited to help learn rich capabilities like depth-aware visual embeddings. Update: my early concerns have been addressed and I would like to raise my score.

---

> ### Author Response · Authors · 2021-08-30
> **Response to Reviewer 13Bf**
>
> Dear Reviewer,
>
> Thanks for your helpful comments and valuable suggestions. We address them as follows.
>
> **Q3.1: Statistically significant results**
>
> A3.1: For all of the experiments (except the robot experiments), we fixed the random seed and parameters, then ran once to get the results. As suggested, we obtain 95% confidence intervals on the test set (via bootstrap), and add these results to the paper. Please see the updated version for the results. In addition, we reran one experiment with 3 different random splits of the training/test data (using ImageNet visual features vs. AV-Order self-supervised visual features, as shown in Table 5). Note that we did this split by *building*, so the difficulty of the task might vary for each trial. The result for each trial is shown below. As in our submission, it can be seen that our model consistently outperforms ImageNet pretrained features.
>
> |Trail | Models   |  Top-1 ACC (%)  |  Top-5 ACC (%)  | Avg Distance  |
> |------|----------|:---------------:|:---------------:|:-------------:|
> |1     | ImageNet | 27.4            | 87.1            | 1.60          |
> |1     | AV-Order | 28.9            | 88.6            | 1.40          |
> |2     | ImageNet | 33.5            | 92.2            | 1.22          |
> |2     | AV-Order | 34.9            | 93.3            | 1.07          |
> |3     | ImageNet | 30.7            | 87.6            | 1.44          |
> |3     | AV-Order | 30.8            | 87.9            | 1.30          |
>
> **Q3.2: Soften the use of language**
>
> A3.2: Thanks for the suggestion. We agree that using "significant" could be interpreted by some as "statistically significant", which was not our intention. We have rephrased all instances of this.

---

### Official Review · Reviewer_8xhP · 2021-07-24

**Originality:** Very Good
**Technical Quality:** Very Good
**Clarity Of Presentation:** Very Good
**Impact:** 4

**Recommendation:**

Weak Accept: I recommend accepting the paper, but will not argue for my recommendation if the majority of other reviewers have a different opinion.

**Summary:**

This work aims to detect obstacles (and walls) from ambient sounds in the environment.  A static dataset is collected for training under two conditions (heating and cooling) and predictions are made both based on human and robotic movements.  Models are evaluated both on absolute and relative depth predictions.  Audio based models are then compared against image based approaches (with and without pretraining) and jointly with them.  Representations are also tested via probe.

**Issues:**

Expanded explanation and discussion of points raised in weaknesses.

**Reviewer Expertise:**

Fair: Some knowledge of the area

**Strengths And Weaknesses:**

**Strengths**
1. Interesting application of acoustic information
2. Construction of novel resource
3. Demonstration of technique with both human and robotic agent
4. Analysis of approach and comparison to and with other modalities

**Weaknesses**
1. Throughout the writing it's unclear where and why a human vs a robot is used.  Specifically, motor noise from the robot itself is not discussed and how this affects the predictions (and/or should be accounted for during training)
2. As best I can tell, all experiments are done within rectangular classroom settings with nearly identical properties/furniture.  This is a worrying simplification.  The walking video does indicate that bags and other random objects may have been present but there's no discussion of their effects.
3. There's a passing note about the sounds of others in the building but no analysis of their effect
4. There's no intuition provided for the shape of the curve in Fig 8.
5. Why in the video (and Fig 6) are predictions never as close to 0 as they are to 1? The gap to zero is (fig 6) 0.18 but the gap to 1 is 0.03.  This appears to be even more dramatic in the supplemental video.

**Summary Of Recommendation:**

This is a very nice (though simple) demonstration of the role of audio.  It does not show a very general formulation nor does it live up to the claims (e.g. refrigerator example) in the introduction. The results are however tested live by both a human and the robotic platform.

---

> ### Author Response · Authors · 2021-08-30
> **Response to Reviewer 8xhP**
>
> Dear Reviewer,
>
> Thanks for your constructive and positive feedback.  We appreciate your acknowledgment of our originality, convincing technical quality and clear structure. We have answered all your great questions below:
>
> **Q2.1: The effect of motor noise**
>
> A2.1: Humans collected all datasets, except for Section 5.4 and the supplement, which evaluated models trained on human-collected data on robot-collected data. When the robots recorded their sounds, they did not move. This was important since it avoided introducing other, non-ambient sounds into the recordings.
>
> We emphasize that recording audio from robots in motion would add *self-produced* sounds. These types of sound have *already been shown to be useful* for navigation through echolocation [11, 12], so it would be unclear which of the two cues (novel ambient sound cues or traditional echo cues) was being used.
>
> To answer the reviewer's question while avoiding self-produced echoes, we added simulated motor noise to the recordings. We collected 31 different servo motor sound effects ([link](https://www.asoundeffect.com/sound-library/servos/)) and synthetically mixed them with our Motion recordings. Audio is therefore corrupted by motor sounds, but they do not provide echo cues. We test under two settings: *Motor-Simple* where the same type of motor noise is added to paired samples, and *Motor-Hard* where every audio sample is mixed with a randomly selected motor noise. We test on the relative depth tasks. The results are shown below. Note that we didn't re-train the model on the synthetic dataset, but the model is still able to generalize well to the noisy data.
>
> | Relative depth   |  AP (%)  |  ACC@0.5 (%)  |
> |---------------|:--------:|:-------------:|
> |Original |    87.1  |      81.3     |
> |Motor-Simple  | 83.0   |  77.9   |
> |Motor-Hard  | 72.0  |  67.4   |
>
> **Q2.2: Similar layouts of classrooms**
>
> A2.2: We feel that our dataset contains a great deal of variation. It contains rooms from 12 different buildings. Their designs vary in floor type (e.g., carpet or hardwood), the presence of windows, wall materials, and furniture, etc. It not only contains classrooms (*Static* recordings), but also conference rooms (*Static* recordings) and hallways (*Motion* recordings). In fact, our dataset is arguably more diverse than many existing related works such as [11, 12, 13].
>
> **Q2.3: The analysis of other sounds in the building**
>
> A2.3: When recording the ambient sound dataset, we tried as much as possible to avoid non-ambient sounds. Although one subset of the dataset was recorded when buildings were occupied, the background noise produced by them is extremely subtle (as they are generally far away). The *Static* recordings, which is the main dataset, were recorded during COVID restrictions when buildings were much less occupied. We provide 100 random samples from *Static* recordings ([link](https://drive.google.com/drive/folders/1K-UbN9NOVm-19AxnVi457xrdkhZApIFR?usp=sharing)) for reviewers to listen. The *Static-dense* recordings were recorded before COVID when the building was occupied, and this is a different type of ambient sound compared with *Static*. Here are random samples ([link](https://drive.google.com/drive/folders/1rza8gIyaE9zYDt6ZXUm7yKuRBQfR-pp6?usp=sharing)) for *Static-dense* recordings.
>
> **Q2.4: Shape of the curve in Figure 8**
>
> A2.4: From Figure 8, we can see, first, that the low-frequency ambient sound is more important than high-frequency for learning scene structure (see Sec. 5.3 for a discussion of the source of these cues). Additionally, note that since we use log-mel spectrograms, the underlying frequency bin size in the x-axis is not uniform. We suspect this is why the curve goes up as the x axis increases: there are more frequencies available to the model (see the x axis labels).
>
> **Q2.5: Unconfident predictions in the demo**
>
> A2.5: In the supplementary video, we tested on a room that was never heard during training, so its prediction is not perfectly well-calibrated. An ideal classifier would be equally confident with both categories. But in practice, when testing on an unseen room, the average response for the classifier varies from room to room.

---

### Official Review · Reviewer_Vy16 · 2021-08-09

**Originality:** Very Good
**Technical Quality:** Very Good
**Clarity Of Presentation:** Excellent
**Impact:** 4

**Recommendation:**

Strong Accept: I recommend accepting the paper and will argue for my recommendation even if other reviewers hold a different opinion.

**Summary:**

* The quiet campus dataset. Large number of in-the-wild indoor ambient audio recordings, paired with concurrently recorded RGB-D data. They collected the data in three modes: static recordings, static with dense coverage and motion recordings.
* Demonstrated audio signals provide a better information for depth estimation over chance.


**Issues:**

* It is not clear whether the audio signal is useful in addition to the visual signal. Training a linear classifier on the depth estimation tasks using both visual and audio features would be a good experiment to evaluate this.

* Train a model with the quiet campus dataset images only and depth estimation as a baseline. This will help discard differences in how the features were trained in ImageNet and also help understand the utility of audio signals.

* It would be interesting to train the robotic navigation agent in the motion recordings. This would be more similar to the kind of data that can be more commonly collected in the wild.

* Testing on a different distribution than training, i.e. training the robotic navigation in the motion recordings and evaluating in a static environment or viceversa.

========================
Update after rebuttal:
All issues have been addressed by the authors.


**Reviewer Expertise:**

Good: General knowledge of the area

**Strengths And Weaknesses:**

Strengths:
* Novel topic.
* Well written and comprehensive examination of related work.
* Thorough experimentation.
* Well presented results. Particularly liked Figure 6.
* The ablation study demonstrating that low frequency audio signals are the most useful for depth estimation is very interesting.

Weaknesses:
* The dataset was collected with the microphone and camera directed towards the closest wall. Since this is the input that is being used to train the models. I worry this is somehow leaking information into the model. This also means the models trained are wouldn't be able to identify depth for any audio signal in the wild, only the ones that were collected in this way.
* The audio-visual representation learning section trains AV-Order and AV-Sync models by using audio + visual tasks. But then, it uses either audio or visual features for training a regression model for obstacle detection and relative order tasks. The audio only models have low performance when compared to the visual ones. The visual models do outperform the pretrained features from Imagenet but a baseline trained on the quiet dataset images only is missing to truly understand the utility of the audio signals.


**Summary Of Recommendation:**

The use of audio in robotics for the purpose of depth estimation and obstacle detection is an appealing idea. It could potentially lead to better ways to avoid collisions during training and deploying navigation policies. However, the paper doesn't argue a clear advantage of using audio signals over visual signals or even in addition to visual signals. Audio signals alone, although better than a random policy, fail to produce performing models (<65% accuracy). The paper doesn't clearly demonstrate the usefulness of using audio signals in addition to visual ones. It is difficult to asses the impact of learning scene structure through audio given that most robotics systems are equipped with a camera these days.

========================
Update after rebuttal:
The experiments show a clear advantage of using audio signals in addition to visual signals. The community would benefit from these results.

---

> ### Author Response · Authors · 2021-08-30
> **Response to Reviewer Vy16 (1/2)**
>
> Dear Reviewer,
>
> Thanks for your thorough review and valuable feedback. We appreciate your acknowledgment of our idea and structure. We have addressed all your concerns below:
>
> **Q1.1: Microphone orientations**
>
> A1.1: It's *not* the case that the microphone was always facing the wall. We evaluated our method on a large number of samples that do not face the wall in the subset of the dataset labeled *Static-dense*. This includes 10h of audio samples with a wide range of angles to the wall (from -90° to 90°). Please refer to L119-L126 for more details about the dataset. Furthermore, our robot experiments show that the audio model shows generalization to other angles, even when trained solely on wall-facing audio.
>
> **Q1.2: The importance of audio cues**
>
> A1.2: There are many advantages to using audio as part of *multimodal* systems: 1) Our new experiments show that using both modalities *together* improves accuracy (see A1.4). 2) One of the promises of multi-modal perception is *robustness to failures*. There are many cases when vision is unavailable, such as when lights in a room are turned off, a lens is occluded, a camera fails, or there is a networking failure.  Audio is an inexpensive signal that is often already available with cameras, and our experiments suggest it could be useful in a "backup" role, much like it is for humans who lose their sense of sight. 3) Audio provides a useful signal for *self-supervised learning* of other modalities. Our experiments suggest that visual models trained with audio are more effective than those trained from scratch (Table 4, Table 5).
>
> **Q1.3: Comparison with a baseline trained on images only**
>
> A1.3: In fact, there are experiments that train an image-based model from scratch in our submission (see Table 1). Compared with linear probing, this end-to-end model obtains worse results on obstacle detection but slightly better on relative depth ordering.
>
> To help understand how audio-visual self-supervision improves *finetuned* models (rather than linear probing), we measured the performance using various numbers of labeled examples. Please see Fig. 16 in our updated supplementary (It can also be seen [here](https://drive.google.com/uc?export=view&id=1iw-8kJjbRUUV-Ls1oqBJ1ccBdJr-b-Sr)). We see a *large improvement from self-supervised initialization* when there is more unlabeled data than labeled data, especially in few-shot training regimes. As expected, when the number of labeled examples equals the number of unlabeled examples in our experiment, the trained-from-scratch model catches up (since the two datasets are exactly the same and the labels provide strictly more information than the audio).
>
> **Q1.4: Linear probing on audio and visual concatenated features**
>
> A1.4: As suggested, we did the linear probing experiment using concatenated visual and audio features, and we compared it with the results of using visual features alone. As shown below, the model with both audio and visual features as inputs performs consistently better than visual features alone, suggesting the audio is useful in addition to the visual signal.
>
> |  Obstacle detection    | Pre.        |  AP (%)  |  ACC@0.5 (%)  |
> |------------------------|:-----------:|:--------:|:-------------:|
> | AV-Order (image)    |             | 76.8     | 68.8          |
> | AV-Order (image + audio)|             | 77.1     | 69.1          |
> | AV-Order (image)    |✓| 86.5| 76.3          |
> | AV-Order (image + audio)|✓| 88.1| 76.9          |
>
> | Relative depth          | Pre.        |  AP (%)  |  ACC@0.5 (%)  |
> |-------------------------|:-----------:|:--------:|:-------------:|
> | AV-Order (image)     |             | 87.4     | 79.1          |
> | AV-Order (image + audio) |             | 89.0     | 80.8          |
> | AV-Order (image)     |✓| 95.8| 88.9          |
> | AV-Order (image + audio) |✓| 95.8| 88.9          |
>
> | Relative depth ratio    | Pre.        |  Top-1 (%)  |  Top-5 (%)  | Avg. Dist     |
> |-------------------------|:-----------:|:-----------:|:-----------:|:-------------:|
> | AV-Order (image)     |             | 24.7        | 80.2        | 1.71          |
> | AV-Order (image + audio) |             | 23.8        | 81.5        | 1.59          |
> | AV-Order (image)     |✓| 28.9        | 88.6        | 1.40          |
> | AV-Order (image + audio) |✓| 30.0        | 89.3        | 1.31          |

---

> > ### Author Response · Authors · 2021-08-30
> > **Response to Reviewer Vy16 (2/2)**
> >
> > **Q1.5: Training the robotic navigation agent with in-the-wild videos**
> >
> > A1.5: We'd like to emphasize that the purpose of the robot experiments is to demonstrate that *ambient* sound is a useful for learning scene structure, and that introducing robotic motion would confound these experiments. For *Motion* recordings, our human operators move slowly to reduce possible sources of sound while recording audio from robots in motion would add *self-produced* sounds. These types of sound have *already been shown to be useful* for navigation through echolocation [11, 12], so it would be unclear which of the two sounds (ambient sounds or echoes) was being used. To answer the reviewer's question while avoiding self-produced echoes, we evaluated adding simulated motor noise to the recordings (see A2.1). These experiments suggest the model performs well despite the presence of these sounds.
> >
> > **Q1.6: Out-of-distribution generalization**
> >
> > A1.6: Our experiments in fact contain a large amount of training/test distribution shift. For example, all the training and testing set is split by room, and the *Motion* recordings is split by building. We also have experiments where models are trained on *Static* subset (recorded in the summer with a single orientation) and tested on *Static-dense* subset (recorded in the winter with multiple orientations), which indicates generalization performance. As suggested, we give the results of the model trained on *Motion* recordings (recorded in hallways) and tested on *Static* recording (recorded in the classrooms). The experiment results are shown as below. Our model can still perform well above random chance despite evaluating on a different domain.
> >
> > |                   |  AP (%)  |  ACC@0.5 (%)  |
> > |-------------------|:--------:|:-------------:|
> > |Obstacle Detection |   61.3   |      57.1     |
> > |Depth Order        |   78.8   |      70.2     |

---

> ### Comment · Reviewer_Vy16 · 2021-09-03
> **Review after Rebuttal**
>
> Thank you for your thorough reply. The experiments show a clear advantage of using audio signals in addition to visual signals. The community would benefit from these results.

---

### Author Response · Authors · 2021-08-30
**Response updated**

We would like to thank all reviewers and the AC for their thorough comments and helpful suggestions. We have replied to each reviewer’s comments individually, and updated the paper and supplementary with corrections, additional experiments, and discussion of limitations.

---

### Meta-Review · Area_Chair_NZKf · 2021-08-13

**Recommendation:** Accept (Oral)
**Confidence:** 4

**Metareview:**

The reviewers generally agree, that the use of acoustics for robot navigation is an interesting concept.
However, the experimental evaluation of the approach is rudimentary and arguably only suffice as a proof of concept.

The reviewers voiced concerns about both, the experimental setup itself and the claims based on the results.
E.g.:
- microphone and camera directed towards the closest wall
- low performance of audio only models
- qualitative statements without reported confidence intervals
- high similarity between spatial layouts of the experiments

The paper should address the concerns of the reviewers wrt the experimental evaluation and add clarifications in the experimental sections.
Furthermore the paper should discuss potential limitations of the work.

## Post-Rebuttal Update:
The authors did a great job clarifying and improving the paper through the rebuttal process.
All major concerns have been addressed.

---

### Decision · Program_Chairs · 2021-09-13

**Decision:**

Accept (Oral)

**Comment:**

The reviewers generally agree, that the use of acoustics for robot navigation is an interesting concept.
However, the experimental evaluation of the approach is rudimentary and arguably only suffice as a proof of concept.

The reviewers voiced concerns about both, the experimental setup itself and the claims based on the results.
E.g.:
- microphone and camera directed towards the closest wall
- low performance of audio only models
- qualitative statements without reported confidence intervals
- high similarity between spatial layouts of the experiments

The paper should address the concerns of the reviewers wrt the experimental evaluation and add clarifications in the experimental sections.
Furthermore the paper should discuss potential limitations of the work.

## Post-Rebuttal Update:
The authors did a great job clarifying and improving the paper through the rebuttal process.
All major concerns have been addressed.